# Implementation strategies to increase human papillomavirus vaccination uptake for adolescent girls in sub-Saharan Africa: A scoping review protocol

**Mwansa Ketty Lubeya**[1,2,3]*, **Mulindi Mwanahamuntu**[1,2], **Carla Chibwesha**[4], **Moses Mukosha**[3,5], **Mercy Monde Wamunyima**[6], **Mary Kawonga**[3]

1 Department of Obstetrics and Gynaecology, School of Medicine, The University of Zambia, Lusaka, Zambia, 2 Women and Newborn Hospital, University Teaching Hospitals, Lusaka, Zambia, 3 School of Public Health, Faculty of Health Sciences, University of the Witwatersrand, Johannesburg, South Africa, 4 Clinical HIV Research Unit, Helen Joseph Hospital, Johannesburg, South Africa, 5 Department of Pharmacy, School of Health Sciences, University of Zambia, Lusaka, Zambia, 6 School of Medicine Library, The University of Zambia, Lusaka, Zambia

* ketty.lubeya@unza.zm

## Abstract

### Introduction

The human papillomavirus (HPV) is sexually transmitted and infects approximately 75% of sexually active people early in their sexual life. Persistent infection with oncogenic HPV types can lead to malignant conditions such as cervical cancer. In 2006, the World Health Organisation approved the use of an efficacious HPV vaccine for girls aged 9 to 14 to prevent HPV-related conditions. Despite the HPV vaccine being available for about 15 years, dose completion remains as low as 20% in sub-Saharan African (SSA) countries implementing the vaccination program compared to 77% in Australia and New Zealand. A fraught of barriers to implementation exist which prevent adequate coverage. Achieving success for HPV vaccination in real-world settings requires strategies to overcome implementation bottlenecks. Therefore, a better understanding and mapping of the implementation strategies used in sub-Saharan Africa to increase HPV vaccination uptake is critical. This review aims to identify implementation strategies to increase HPV vaccination uptake for adolescent girls in sub-Saharan Africa and provide a basis for policy and future research, including systematic reviews to evaluate effective strategies as we accelerate the elimination of cervical cancer.

### Materials and methods

This scoping review will consider studies pertaining to implementation strategies to increase HPV vaccination uptake for adolescent girls in sub-Saharan Africa. Studies targeted at different stakeholders to increase adolescent vaccine uptake will be included. Studies using interventions not fitting the definition of implementation strategies as defined by the refined compilation of implementation strategies from the Expert Recommendations for

**Data Availability Statement:** All relevant data are within the article and its Supporting Information files.

**Funding:** Research funding and trainee support for MKL was provided by the UNC-UNZA-Wits Partnership for HIV and Women's Reproductive Health which is funded by the US National Institute's Health (grant number D43 TW010558). The funders had and will not have a role in study design, data collection and analysis, decision to publish, or preparation of the manuscript.

**Competing interests:** The authors have declared that no competing interests exist

Implementing Change project will be excluded. MEDLINE (via PubMed), Embase, CINAHL (via EBSCO), Scopus and Google Scholar will be searched. Two independent reviewers will screen titles and abstracts for studies that meet the review's inclusion criteria, and the full text of eligible studies will be reviewed. Data will be extracted from eligible studies using a structured data charting table developed by this team for inclusion by two independent reviewers and presented in a table and graphical form with a narrative summary.

## Introduction

The human papillomavirus (HPV) is sexually transmitted and affects approximately 75% of sexually active people early in their sexual life [1, 2]. There are more than 200 species of HPVs categorized into low-risk (causing warts) and high-risk types (causing cancers) [3]. The point prevalence of HPV infection is estimated at 11–12% globally and 22–24% for sub-Sahara Africa (SSA) [4–7]. HPV infection is mainly asymptomatic and transient, with approximately 90% cleared in immunocompetent people but persists in about 10–20% [2]. Immunocompromised people, such as those living with HIV, are more likely to have persistent HPV infections, which usually progress more rapidly to pre-cancers and cancers than the general population [8]. Cancers caused by HPV include vaginal cancer, vulvar cancer, and anal cancer, as well as some head and neck cancers like oropharyngeal cancer and cervical cancer which is highly prevalent in SSA [5].

In 2006, the World Health Organisation (WHO) approved HPV vaccines for prepubertal girls aged 9 to 14 years, naïve to sexual intercourse to prevent HPV-related conditions [2]. There are currently four licensed HPV vaccines classified as bivalent, quadrivalent and nonavalent based on the number of targeted HPV strains [9]. From 2006 to 2014, three doses were recommended to complete the vaccination schedule; however, evolving evidence has shown that two doses are effective, therefore, WHO recommends two doses for girls below the age of 15 [10], with ongoing studies showing promise for an efficacious and cost-effective single-dose HPV vaccine [11]. Therefore, the three-dose schedule is now reserved for girls older than 15 years or the immunocompromised, including people living with HIV [9]. The reduced number of doses is envisioned to boost vaccine coverage and reduce barriers to vaccination, such as multiple visits to the health facility, human resource demand and vaccine cost [12–15].

Despite the HPV vaccine being available for about 15 years, dose completion remains as low as 20% in SSA for countries implementing the vaccination program compared to 77% in Australia and New Zealand [16]. Barriers to HPV vaccination have been identified within SSA, and implementing national programs remains challenging at both initiation and sustainability levels [17]. Some identified barriers to HPV vaccine uptake include; individual, structural, economic, community/social, and cultural issues [14, 18]. Further, systematic health system constraints occur in many SSA countries, including but not limited to service delivery, low levels of knowledge among health care workers, financing, vaccine communication, and community engagement [19]. There is a dire need at the global level to increase HPV vaccine uptake with the accelerated global strategy for cervical cancer elimination, whose target is to fully vaccinate 90% of girls aged 15 years by 2030 [20].

The impact of HPV vaccination in reducing HPV-related diseases is evident in high-income countries which introduced national immunization programs earlier. For example, a meta-analysis of fourteen high-income countries implementing the HPV vaccination program reported a reduction in the prevalence of HPV 16 and 18 by 83% among girls aged 13–19 years and by

66% among women aged 20–24 years with significant cross-protection from HPV 31, 33 and 45 [21]. Furthermore, emerging data from the UK has shown that cervical cancer has almost been eliminated ten years after implementing the HPV vaccination of adolescent girls aged 12–13 years [22]. However, many SSA countries are still trailing behind in implementing the national HPV vaccination under a backdrop of low uptake in countries that have already implemented the program despite having the highest HPV-related morbidity and mortality [12, 23].

The point of the HPV vaccination programme is to get vaccines to those who need to receive them and mitigate future preventable diseases such as cervical cancer. For interventions such as the HPV vaccination to achieve success (intervention effectiveness) in a real-world setting, they must be implemented well and in ways that can reach the target groups. This often requires implementation strategies to overcome any implementation bottlenecks [24]. According to Powel and others, an implementation strategy is "a systematic intervention process to adopt and integrate evidence-based health innovations into usual care" [24]. Therefore, to successfully implement the HPV vaccine and harness the benefits, implementation strategies should be tailored to a real-life situation. For example, a study by Chigbu et al. [25] in Nigeria used educational intervention [implementation strategy] to increase HPV vaccine uptake for adolescent girls from 0.9% to 33.2% in a pre and post-design method. Another study done in South Africa used multiple implementation strategies such as building a coalition, conducting educational meetings, conducting educational outreach visits, developing educational materials, distributing educational materials, mandate change, obtaining and using consumers feedback and saw adolescent HPV vaccine uptake of up to 87% [26]. A systematic review by Niccolai et al. [27], including studies from the USA, reported implementation strategies such as reminder and recall systems, physician-focused strategies, school-located programs, and social marketing to have improved HPV vaccination coverage among adolescents. Another systematic review focused on high-income countries identified reminders, education and training, information delivery and communication campaigns, feedback on coverage data, financial incentives and school-based interventions as strategies which increased adolescent HPV vaccination [28].

There is a consensus among researchers that implementation strategies in one region may not necessarily work in another; therefore, implementation strategies should be tailored according to the context in which they will be implemented in a real-life context [24]. Therefore, it is urgent to have evidence from SSA, one of the regions with the highest burden of HPV infection and align with the global strategy for eliminating cervical cancer [20]. Therefore, this scoping review aims to identify implementation strategies used in SSA to increase HPV vaccine uptake for adolescent girls. The findings of this review will provide systematic evidence and an opportunity for policymakers to link the identified implementation strategies to the systematic barriers to HPV vaccination implementation. In addition, the findings will form a basis for future research, including systematic reviews to understand which implementation strategies are effective in SSA in tandem with the recommendations by the WHO Africa region leadership [17].

A preliminary literature review was conducted in December 2021 inPubMed, Scopus, Google Scholar, and JBI evidence synthesis database; no similar scoping review has been done or is underway in the context of SSA. Even though a narrative review was conducted on the HPV vaccine in SSA in 2020, focusing on ten years of HPV vaccination research in Tanzania [29]. However, a narrative review uses an implicit process to compile evidence and has weaknesses such as lack of systematic selection of studies and likely to reinforce authors' preconceived ideas and experiences, leading to bias [30]. On the other hand, our scoping review will be comprehensive with explicit inclusion and exclusion criteria and will transparently synthesise the available evidence to minimize similar limitations.

### Review question

What implementation strategies have been used to increase the uptake of HPV vaccination for adolescent girls in sub-Saharan Africa?

### Objective

To identify implementation strategies used to increase HPV vaccination uptake for adolescent girls in sub-Saharan Africa and provide a basis for future systematic reviews to evaluate effective strategies.

## Materials and methods

The proposed scoping review will follow the updated Joanna Briggs Institute (JBI) methodology for scoping reviews [31]. This scoping review is registered in Open Science Framework projects [32].

### Inclusion criteria

**Participants.** Participants will include all stakeholders involved in the cascade of HPV vaccination, such as studies on adolescent girls aged 9–14 years as the primary recipients and those aged 15–19 years who may be eligible for subsequent doses. This scoping review will also include studies focusing on parents/guardians of adolescent girls as they are involved in the consent process for the vaccination of their daughters. Further, studies involving healthcare workers will also be included considering that healthcare providers play a crucial role in recommending and administering the vaccine to eligible adolescent girls. Finally, studies involving teachers and other community leaders will also be considered.

**Concept.** We will include studies that used at least one of the implementation strategies defined by the refined compilation of implementation strategies from the Expert Recommendations for Implementing Change (ERIC) project [33] to increase the uptake of HPV vaccination. The ERIC expert panel achieved consensus on 73 discrete implementation strategies, including their definitions. Some examples of these implementations strategies include; i) conduct educational meetings, defined as "hold meetings targeted toward different stakeholder groups to teach them about the clinical innovation," ii) build a coalition, defined as "recruit and cultivate relationships with partners in the implementation effort," iii) access new funding defined as "access new or existing money to facilitate the implementation" [33]. The ERIC project classification has been used in scoping reviews in other health fields such as dementia [34, 35] and costing of implementation strategies [36] scoping reviews in preference to other taxonomies as it is more detailed with a conceptually clear description of strategies.

Considering that not all interventions may be reported as implementation strategies, we will read through all the studies mentioning interventions to check which ones meet the definitions by the ERIC project and classify them during data charting. Studies including interventions other than those based on the refined compilation of implementation strategies from the ERIC project will be excluded.

**Context.** This review will consider studies done in sub-Saharan Africa aimed at improving the uptake of the HPV vaccine among adolescent girls.

### Types of sources

This scoping review will consider experimental and quasi-experimental study designs, including randomized controlled trials, non-randomized controlled trials, before and after studies and interrupted time-series studies. In addition, analytical observational studies, including

prospective and retrospective cohort studies, case-control studies and analytical cross-sectional studies, will be considered for inclusion. We will also consider descriptive observational study designs and qualitative studies. In addition, systematic reviews that meet the inclusion criteria will also be considered, depending on the research question. Text and opinion papers will also be considered.

## Search strategy

An initial limited search of MEDLINE (PubMed), EMBASE and Scopus was undertaken to identify articles on the topic. The text words contained in the titles and abstracts of relevant articles and the index terms used to describe the articles were used to develop a full search strategy for MEDLINE (PubMed) (S1 Appendix). The search strategy will aim to locate both published and unpublished studies. The search strategy, including all identified keywords and index terms, will be adapted for each included database and information source. The reference list of included sources of evidence will be screened for additional studies. The section for similar articles will also be searched on MEDLINE (PubMed).

Studies published in any other language besides English will not be included as the teams' competence does not go beyond the English language. Studies published starting in 2006 will be included as this was when WHO approved the use of the HPV vaccine among prepubertal girls; the search will include studies published until December 31$^{st}$ 2021.

The databases to be searched include MEDLINE (via PubMed), EMBASE, CINAHL (Cumulative Index to Nursing and Allied Health Literature) (EBSCO) and Scopus. Grey literature citations (unpublished, nonpeer review data sources including conference proceedings, abstracts and dissertations) indexed by any databases searched will be reviewed with other peer-reviewed publications using the same inclusion and exclusion criteria. Google Scholar is useful, particularly in grey literature search, despite its limitations. Systematic reviews have traditionally assessed the first 50 search results from Google Scholar to complement searches from other databases [37, 38]. Therefore, we will review the first 50 hits from the Google Scholar search engine for possible inclusion. A medical librarian/ information specialist will guide the search.

Following the search, all identified studies will be collated and uploaded into Endnote X9 (Clarivate Analytics, USA) and duplicates removed, after which they will be imported into the Joanna Briggs Institute System for the Unified Management, Assessment and Review of Information (JBI SUMARI) [39]. Two independent reviewers will screen titles and abstracts for assessment against the inclusion criteria for the review. Potentially relevant studies will be retrieved in full, and their citation details using JBI SUMARI. Two independent reviewers will assess the full text of selected citations in detail against the inclusion criteria. The scoping review will record and report reasons for excluding full-text studies that do not meet the inclusion criteria. Any disagreements that arise between the reviewers at each stage of the study selection process will be resolved through discussion or with a third reviewer if consensus is not reached. The search is planned to start on 31$^{st}$ May 2022. The search results will be reported in full in the final scoping review and presented in a Preferred Reporting Items for Systematic Reviews and Meta-analyses extension for scoping reviews (PRISMA-ScR) flow diagram [40].

## Data extraction

Data will be extracted from papers included in the scoping review by two independent reviewers using a data extraction chart developed by the reviewers (S2 Appendix). The data extracted will include; the first author, year of study, year of publication, country, the title of the article,

article type, study design (where applicable), sample size, implementation strategy(ies), targeted stakeholders, type of program (national program/demonstration/subpopulation, the funding source for the vaccine, HPV vaccine coverage/uptake, limitations and strengths. The draft extraction chart will be piloted on five included pieces of evidence and revisions made if necessary. The draft data extraction tool will further be modified and revised as necessary during data charting from each included evidence source. Modifications will be detailed in the scoping review. Authors of included papers will be contacted to request missing or additional data, where required; if no response is received within two weeks, available data will be used in the best possible way.

## Data presentation

The scoping review reporting will align with the PRISMA-ScR [40]. The extracted data will be presented in a table or graphical form to align with the study objectives. These will accompany a narrative summary of how the findings relate to the research question and objectives.

## Discussion

A narrative summary of findings will be provided and discussed.

## Limitations

Firstly, this scoping review will only focus on studies done in English; hence may miss other important research conducted in other languages. Secondly, the expert panel involved in developing the ERIC taxonomy was drawn from the USA and North America; hence some strategies may be more applicable to North American settings. However, the original compilation drew from taxonomies developed in other contexts [33]. Thirdly, some studies may not explicitly define the interventions as implementation strategies, however, this will be mitigated by a thorough review of all identified interventions against the definitions of implementation strategies by the ERIC project. Finally, this scoping review will not include studies that examined interventions in girls who are 20 years or older.

## Supporting information

**S1 Appendix. PUBMED search.**
(DOCX)

**S2 Appendix. Sample data charting tool.**
(DOCX)

**S3 Appendix. PRISMA ScR checklist.**
(DOCX)

## Acknowledgments

We want to acknowledge the support from the University of Zambia-University of North Carolina at ChapelHill and Witwatersrand University consortium PhD fellowship program.

## Author Contributions

**Conceptualization:** Mwansa Ketty Lubeya.

**Funding acquisition:** Mwansa Ketty Lubeya, Carla Chibwesha.

**Methodology:** Mwansa Ketty Lubeya, Mulindi Mwanahamuntu, Carla Chibwesha, Moses Mukosha, Mercy Monde Wamunyima, Mary Kawonga.

**Resources:** Mwansa Ketty Lubeya, Mercy Monde Wamunyima.

**Supervision:** Mulindi Mwanahamuntu, Carla Chibwesha, Moses Mukosha, Mary Kawonga.

**Writing – original draft:** Mwansa Ketty Lubeya.

**Writing – review & editing:** Mwansa Ketty Lubeya, Mulindi Mwanahamuntu, Carla Chibwesha, Moses Mukosha, Mercy Monde Wamunyima, Mary Kawonga.

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
