## [Decision Letter · Decision Letter 0]

27 Jun 2022

PONE-D-22-10538Implementation strategies to increase human papillomavirus vaccination uptake for adolescent girls in sub-Saharan Africa: A scoping review protocolPLOS ONE

Dear Dr. Lubeya,

Thank you for submitting your manuscript to PLOS ONE. After careful consideration, we feel that it has merit and we are happy to accept the manuscript for publication pending minor revision. Therefore, we invite you to submit a revised version of the manuscript that addresses the points raised during the review process.

The protocol publication is recommended, but before, the authors should address a minor revision as suggested in the reviewers' comments and suggestions below. Please address the reviewers' comments and suggestions and submit the revised version of the protocol in the next two weeks.

REVIEWER #1:

*Recognizing the need to scale-up Human Papillomavirus (HPV) vaccination in sub-Saharan Africa where the disease burden and need for intervention are among the highest in the world, this proposed scoping review intends to map available evidence on implementation strategies aimed at increasing HPV vaccine uptake among eligible adolescent girls*. *Given the dearth of evidence from this setting, the findings of this proposed review will be useful for informing interventions to improve HPV vaccine acceptance and uptake. Overall, the proposal is well written save for minor editorial/grammatical errors. The following comments need addressing:*

Major Comments

Given that challenges with achieving optimal HPV vaccine uptake or dose completion rates are not unique to sub-Saharan Africa, could the authors further support the argumentation presented here by appraising evidence on implementation strategies used to improve vaccine uptake among pre-adolescent and adolescent girls in countries external to sub-Saharan Africa?

The review question suggests that the authors will only be reviewing published / unpublished evidence of proven effectiveness, i.e., where implementation strategies have in fact been shown to improve HPV vaccine uptake, and therefore excluding evidence on those systematic interventions that are yet to show improvements in vaccine uptake. Is this truly the case, or will the scoping review map all available evidence on implementation strategies explored within the sub-Saharan African context with the primary aim of improving HPV vaccine uptake?

With regards to the Concept, the authors may want to buttress their selection of the ERIC project definition of implementation strategies by providing and citing previous reviews that have used / validated the refined compilation of implementation strategies recommended, including reviews / studies focused on the sub-Saharan African region. Furthermore, in the Limitations section of this manuscript, could the authors comment on any foreseeable limitations with using the EPIC project definition of implementation strategies.

Minor Comments

Line 66, page 3; In addition to the framework for elimination of cervical cancer within the WHO Afro region, the authors should more specifically cite appropriate systematic evidence on the barriers to HPV vaccination within sub-Saharan Africa.

Line 71, page 3; “…whose target is to have 90% of girls aged 15 should be fully vaccinated by 2030” Kindly rephrase for clarity - remove "should be".

Line 103, page 4; “…The findings of this review will provide a basis for policy…” The authors should explicitly address how the anticipated findings of this proposed scoping review could inform current policy / policy reform on HPV vaccination in sub-Saharan Africa.

Lines 132 – 133, page 5; “…healthcare providers play a crucial role in recommending and administering the vaccine to eligible.” This sentence is incomplete.

Lines 142 – 143, page 6; “…iii) access new funding defined as “access new or existing money to facilitate the implementation” [25].” The authors should confirm that they have cited the corrected reference here as reference number 25 does not relate to the ERIC project.

Lines 203 – 205, page 8; “Authors of papers will be contacted … if no response will be received within two weeks, available data will be used in the best possible way.” Kindly rephrase tense for ease of clarity.

Lines 217 – 218, page 8; “…could be other unique and successful interventions used to increase HPV vaccine uptake among boys.” Could the authors expand by citing references on HPV vaccine uptake among boys in sub-Saharan African countries?

Lines 225 – 226, page 9; “…MKL: Conceptualised and drafted the initial manuscript, MMW & conducted the pilot…” This sentence appears incomplete.

*Supporting information S2: Sample Data Charting Form - I’d suggest that the authors consider splitting the table to enhance the presentation and clarity of all the data extraction variables*.

REVIEWER #2:

*This is a well done scoping review protocol*. *The authors have done an excellent job of justifying the need for the review and presenting the methodology. I have only three minor suggestions to consider, and none of these are critical:*

1. Add scoping review as a keyword

2. Provide a justification or citation for only reviewing the first 50 hits from Google Scholar. This is not a reproducible strategy so I question whether it is even needed.

3. Add as a limitation that this review will not include any studies that examined interventions in 20+ year olds.

*Thank you for your work* - *this was an easy to read manuscript!*

We look forward to receiving your revised manuscript.

Kind regards,

Ana Catarina Canário

Academic Editor

PLOS ONE

Journal Requirements:

Additional Editor Comments:

Dear Ms. Mwansa Ketty Lubeya,

Thank you very much for considering PLOS ONE for submitting your scoping review protocol: "Implementation strategies to increase human papillomavirus vaccination uptake for adolescent girls in sub-Saharan Africa: A scoping review protocol". Two reviewers have read your manuscript and found it relevant and well-written. This scoping review protocol is clear and elucidative of the work the authors propose to complete in the scoping review, rigorously detailing the procedures that the authors will use in their scoping review.

The protocol publication is recommended, but before, the authors should address a minor revision as suggested in the reviewers' comments and suggestions below. Please address the reviewers' comments and suggestions and submit the revised version of the protocol in the next two weeks, until July 5th, 2022.

Once again, thank you for your contribution to PLOS ONE!

Reviewers' comments:

Reviewer's Responses to Questions

**Comments to the Author**

1. Does the manuscript provide a valid rationale for the proposed study, with clearly identified and justified research questions?

Reviewer #1: Yes

Reviewer #2: Yes

2. Is the protocol technically sound and planned in a manner that will lead to a meaningful outcome and allow testing the stated hypotheses?

Reviewer #1: Yes

Reviewer #2: Yes

3. Is the methodology feasible and described in sufficient detail to allow the work to be replicable?

Reviewer #1: Yes

Reviewer #2: Yes

4. Have the authors described where all data underlying the findings will be made available when the study is complete?

Reviewer #1: Yes

Reviewer #2: Yes

5. Is the manuscript presented in an intelligible fashion and written in standard English?

Reviewer #1: Yes

Reviewer #2: Yes

6. Review Comments to the Author

You may also provide optional suggestions and comments to authors that they might find helpful in planning their study.

Reviewer #1: Recognizing the need to scale-up Human Papillomavirus (HPV) vaccination in sub-Saharan Africa where the disease burden and need for intervention are among the highest in the world, this proposed scoping review intends to map available evidence on implementation strategies aimed at increasing HPV vaccine uptake among eligible adolescent girls. Given the dearth of evidence from this setting, the findings of this proposed review will be useful for informing interventions to improve HPV vaccine acceptance and uptake. Overall, the proposal is well written save for minor editorial/grammatical errors. The following comments need addressing:

Major Comments

Given that challenges with achieving optimal HPV vaccine uptake or dose completion rates are not unique to sub-Saharan Africa, could the authors further support the argumentation presented here by appraising evidence on implementation strategies used to improve vaccine uptake among pre-adolescent and adolescent girls in countries external to sub-Saharan Africa?

The review question suggests that the authors will only be reviewing published / unpublished evidence of proven effectiveness, i.e., where implementation strategies have in fact been shown to improve HPV vaccine uptake, and therefore excluding evidence on those systematic interventions that are yet to show improvements in vaccine uptake. Is this truly the case, or will the scoping review map all available evidence on implementation strategies explored within the sub-Saharan African context with the primary aim of improving HPV vaccine uptake?

With regards to the Concept, the authors may want to buttress their selection of the ERIC project definition of implementation strategies by providing and citing previous reviews that have used / validated the refined compilation of implementation strategies recommended, including reviews / studies focused on the sub-Saharan African region. Furthermore, in the Limitations section of this manuscript, could the authors comment on any foreseeable limitations with using the EPIC project definition of implementation strategies.

Minor Comments

Line 66, page 3; In addition to the framework for elimination of cervical cancer within the WHO Afro region, the authors should more specifically cite appropriate systematic evidence on the barriers to HPV vaccination within sub-Saharan Africa.

Line 71, page 3; “…whose target is to have 90% of girls aged 15 should be fully vaccinated by 2030” Kindly rephrase for clarity - remove "should be".

Line 103, page 4; “…The findings of this review will provide a basis for policy…” The authors should explicitly address how the anticipated findings of this proposed scoping review could inform current policy / policy reform on HPV vaccination in sub-Saharan Africa.

Lines 132 – 133, page 5; “…healthcare providers play a crucial role in recommending and administering the vaccine to eligible.” This sentence is incomplete.

Lines 142 – 143, page 6; “…iii) access new funding defined as “access new or existing money to facilitate the implementation” [25].” The authors should confirm that they have cited the corrected reference here as reference number 25 does not relate to the ERIC project.

Lines 203 – 205, page 8; “Authors of papers will be contacted … if no response will be received within two weeks, available data will be used in the best possible way.” Kindly rephrase tense for ease of clarity.

Lines 217 – 218, page 8; “…could be other unique and successful interventions used to increase HPV vaccine uptake among boys.” Could the authors expand by citing references on HPV vaccine uptake among boys in sub-Saharan African countries?

Lines 225 – 226, page 9; “…MKL: Conceptualised and drafted the initial manuscript, MMW & conducted the pilot…” This sentence appears incomplete.

Supporting information S2: Sample Data Charting Form - I’d suggest that the authors consider splitting the table to enhance the presentation and clarity of all the data extraction variables.

Reviewer #2: This is a well done scoping review protocol. The authors have done an excellent job of justifying the need for the review and presenting the methodology. I have only three minor suggestions to consider, and none of these are critical:

1. Add scoping review as a keyword

2. Provide a justification or citation for only reviewing the first 50 hits from Google Scholar. This is not a reproducible strategy so I question whether it is even needed.

3. Add as a limitation that this review will not include any studies that examined interventions in 20+ year olds.

Thank you for your work - this was an easy to read manuscript!

7. PLOS authors have the option to publish the peer review history of their article (what does this mean?). If published, this will include your full peer review and any attached files.

Reviewer #1: **Yes: **Edina Amponsah-Dacosta

Reviewer #2: No

---

## [Author Response · Author response to Decision Letter 0]

19 Jul 2022

Thank you, the responses to the reviewers have been attached to the rebuttal letter

---

## [Decision Letter · Decision Letter 1]

8 Aug 2022

Implementation strategies to increase human papillomavirus vaccination uptake for adolescent girls in sub-Saharan Africa: A scoping review protocol

PONE-D-22-10538R1

Dear Dr. Lubeya,

We’re pleased to inform you that your manuscript has been judged scientifically suitable for publication and will be formally accepted for publication once it meets all outstanding technical requirements.

Within one week, you’ll receive an e-mail detailing the required amendments. When these have been addressed, you’ll receive a formal acceptance letter and your manuscript will be scheduled for publication. Please address the following aspects in your manuscript before publication, per recommendation of reviewer#1:

1) Please proofread the manuscript and address minor grammatical errors.

2) Please ensure that both abstracts provided in this manuscript are the same. For example, the abstract on page 1 indicates the following, “…dose completion remains at 53% in sub-Saharan Africa for countries implementing the vaccination program”. The abstract on page 7 (Lines 21 – 22) however, reads as follows, "...dose completion remains as low as 20% in sub-Saharan African (SSA) countries implementing the vaccination program compared to 77% in Australia and New Zealand."

3) Please clarify the information in Lines 52 – 53; “Cancers caused by HPV include the anal-genital and the head and neck regions…” This statement is unclear. The authors should either list the cancers associated with persistent HPV infection or rephrase the sentence to indicate that persistent HPV infection has been associated with the development of cancers in the specific regions listed (e.g., "Cancers caused by HPV include vaginal cancer, vulvar cancer, and anal cancer, as well as some head and neck cancers like oropharyngeal cancer"). Further to this, given the focus on cervical cancer prevention within sub-Saharan Africa, I'd urge the authors to explicitly mention this form of cancer as part of this sentence.

Kind regards,

Ana Catarina Miranda Canário

Academic Editor

PLOS ONE

Additional Editor Comments:

Dear Dr. Lubeya,

Thank you very much for considering PLOS ONE for submitting your scoping review protocol: "Implementation strategies to increase human papillomavirus vaccination uptake for adolescent girls in sub-Saharan Africa: A scoping review protocol". Two reviewers have read your manuscript and considered that you addressed their prior comments and suggestions. Your manuscript is now accepted for publication in PLOS ONE.

However, I would like to draw your attention to the comments made by reviewer #1, as they recommend that before publication:

1) The manuscript is proofread and minor grammatical errors addressed.

2) Please ensure that both abstracts provided in this manuscript are the same. For example, the abstract on page 1 indicates the following, “…dose completion remains at 53% in sub-Saharan Africa for countries implementing the vaccination program”. The abstract on page 7 (Lines 21 – 22) however, reads as follows, "...dose completion remains as low as 20% in sub-Saharan African (SSA) countries implementing the vaccination program compared to 77% in Australia and New Zealand."

3) Please clarify the information in Lines 52 – 53; “Cancers caused by HPV include the anal-genital and the head and neck regions…” This statement is unclear. The authors should either list the cancers associated with persistent HPV infection or rephrase the sentence to indicate that persistent HPV infection has been associated with the development of cancers in the specific regions listed (e.g., "Cancers caused by HPV include vaginal cancer, vulvar cancer, and anal cancer, as well as some head and neck cancers like oropharyngeal cancer"). Further to this, given the focus on cervical cancer prevention within sub-Saharan Africa, I'd urge the authors to explicitly mention this form of cancer as part of this sentence.

Thank you for your contribution to PLOS ONE!

Reviewers' comments:

Reviewer's Responses to Questions

**Comments to the Author**

1. Does the manuscript provide a valid rationale for the proposed study, with clearly identified and justified research questions?

Reviewer #1: Yes

Reviewer #2: Yes

2. Is the protocol technically sound and planned in a manner that will lead to a meaningful outcome and allow testing the stated hypotheses?

Reviewer #1: Yes

Reviewer #2: Yes

3. Is the methodology feasible and described in sufficient detail to allow the work to be replicable?

Reviewer #1: Yes

Reviewer #2: Yes

4. Have the authors described where all data underlying the findings will be made available when the study is complete?

Reviewer #1: Yes

Reviewer #2: Yes

5. Is the manuscript presented in an intelligible fashion and written in standard English?

Reviewer #1: Yes

Reviewer #2: Yes

6. Review Comments to the Author

You may also provide optional suggestions and comments to authors that they might find helpful in planning their study.

Reviewer #1: The authors have adequately addressed the comments raised during the initial phase of peer review. The manuscript is much improved. However, minor grammatical / editorial errors require the authors’ attention before being finalized.

Minor Comments:

Kindly ensure that both abstracts provided in this manuscript are the same. For example, the abstract on page 1 indicates the following, “…dose completion remains at 53% in sub-Saharan Africa for countries implementing the vaccination program”. The abstract on page 7 (Lines 21 – 22) however, reads as follows, "...dose completion remains as low as 20% in sub-Saharan African (SSA) countries implementing the vaccination program compared to 77% in Australia and Newzealand."

Lines 52 – 53; “Cancers caused by HPV include the anal-genital and the head and neck regions…” This statement is unclear. The authors should either list the cancers associated with persistent HPV infection or rephrase the sentence to indicate that persistent HPV infection has been associated with the development of cancers in the specific regions listed (e.g., "Cancers caused by HPV include vaginal cancer, vulvar cancer, and anal cancer, as well as some head and neck cancers like oropharyngeal cancer"). Further to this, given the focus on cervical cancer prevention within sub-Saharan Africa, I'd urge the authors to explicitly mention this form of cancer as part of this sentence.

Reviewer #2: The authors have addressed all of the review feedback well. I have no further suggested edits. I recommend that the manuscript is ready for publication.

7. PLOS authors have the option to publish the peer review history of their article (what does this mean?). If published, this will include your full peer review and any attached files.

Reviewer #1: **Yes: **Edina Amponsah-Dacosta

Reviewer #2: No

---

## [Editor Report · Acceptance letter]

16 Aug 2022

PONE-D-22-10538R1 

Implementation strategies to increase human papillomavirus vaccination uptake for adolescent girls in sub-Saharan Africa: A scoping review protocol 

Dear Dr. Lubeya:

I'm pleased to inform you that your manuscript has been deemed suitable for publication in PLOS ONE. Congratulations! Your manuscript is now with our production department. 

Kind regards, 

on behalf of

Dr. Ana Catarina Miranda Canário 

Academic Editor

PLOS ONE